# Crystal structures of Rea1-MIDAS bound to its ribosome assembly factor ligands resembling integrin–ligand-type complexes

Yasar Luqman Ahmed [1,3], Matthias Thoms[1,2,3], Valentin Mitterer [1], Irmgard Sinning [1] & Ed Hurt [1]

The Rea1 $AAA^+$-ATPase dislodges assembly factors from pre-60S ribosomes upon ATP hydrolysis, thereby driving ribosome biogenesis. Here, we present crystal structures of Rea1-MIDAS, the conserved domain at the tip of the flexible Rea1 tail, alone and in complex with its substrate ligands, the UBL domains of Rsa4 or Ytm1. These complexes have structural similarity to integrin α-subunit domains when bound to extracellular matrix ligands, which for integrin biology is a key determinant for force-bearing cell–cell adhesion. However, the presence of additional motifs equips Rea1-MIDAS for its tasks in ribosome maturation. One loop insert cofunctions as an NLS and to activate the mechanochemical Rea1 cycle, whereas an additional β-hairpin provides an anchor to hold the ligand UBL domains in place. Our data show the versatility of the MIDAS fold for mechanical force transmission in processes as varied as integrin-mediated cell adhesion and mechanochemical removal of assembly factors from pre-ribosomes.

[1] Heidelberg University Biochemistry Center, D-69120 Heidelberg, Germany. [2] Present address: Gene Center, University of Munich, D-81377 Munich, Germany. [3] These authors contributed equally: Yasar Luqman Ahmed, Matthias Thoms. Correspondence and requests for materials should be addressed to I.S. (email: irmi.sinning@bzh.uni-heidelberg.de) or to E.H. (email: ed.hurt@bzh.uni-heidelberg.de)

Ribosomes are nanomachines composed of ribosomal RNA (5S, 5.8S, 28/25S, and 18S rRNA) and ~80 ribosomal proteins, which, together with other factors, aminoacylated tRNAs, and mRNAs, synthesize proteins in the cytoplasm. Ribosomes themselves are synthesized and assembled in the nuclear compartment along a sophisticated assembly line with many modification, processing and maturation steps, before export to the cytoplasm. Ribosome synthesis in growing cells requires a vast amount of energy and is tightly regulated and coordinated with other cellular pathways[1]. It is therefore not surprising that several diseases have been linked to ribosome assembly disorders, and strong upregulation of rRNA synthesis and ribosome assembly is observed during cancer development[2–5].

Ribosome biogenesis starts with transcription of a precursor rRNA (called 35S and 47S pre-rRNA in yeast and humans, respectively) in the nucleolus, from which mature 18S, 5.8S, and 25S (28S in humans) rRNAs are generated through a series of RNA processing reactions[6,7]. During pre-rRNA transcription, ribosomal proteins, and early biogenesis factors associate with the nascent rRNA to form the first pre-ribosomal particles, called 90S pre-ribosomes or small-subunit processomes and for which pseudo-atomic structures were published recently[8–11]. Endonucleolytic cleavage of the 35S/47S pre-rRNA separates the maturation pathways of the small (40S) and large (60S) subunit. Whereas the 40S precursor particle is rapidly exported to the cytoplasm where final maturation steps occur[12–16], the recruitment of ribosomal proteins together with a plethora of ribosome biogenesis factors leads to the formation of the earliest pre-60S particles in the nucleolus[17,18]. These 60S precursors undergo a series of re-arrangement and quality-control steps on their way from the nucleolus through the nucleoplasm, before they are finally exported to the cytoplasm[19–26]. During these maturation steps, particular assembly factors with ATPase or GTPase activities give the pre-60S maturation pathway directionality[27].

One of these energy-consuming enzymes is the conserved huge AAA$^+$-type ATPase Rea1 (also known as Mdn1), which is composed of a hexameric AAA$^+$-ATPase domain, followed by a long tail consisting of an α-helical linker region, a flexible aspartate/glutamate (D/E)-rich domain and a C-terminal metal-ion-dependent adhesion site (MIDAS) homologous to the inserted (I) domain within the α-subunits (αI) of integrins that mediate ligand recognition[28–34]. Crystal structures of MIDAS domains from the integrin receptor in complex with its ligand showed that the divalent metal ion ($Mg^{2+}$) at the integrin–ligand interface is coordinated by five conserved residues of the MIDAS fold (consensus $DXSXS\text{-}X_{70}\text{-}T\text{-}X_{30}\text{-}(S/T)DG$) and the sixth coordinating residue (either E or D) is provided by the ligand[35,36]. This interaction is key in the mechanical interactions between cells and the extracellular matrix and plays a central role in directing cell migration and proliferation. The conserved interaction between the C-terminal Rea1-MIDAS and the Rsa4-UBL (ubiquitin-like domain) or Ytm1-UBL is essential for assembly of the 60S pre-ribosome in a reaction mediated by the Rea1 ATPase upon ATP hydrolysis that leads to the removal of Ytm1 and Rsa4 from distinct pre-60S intermediates[25,37,38]. Essential for binding to the Rea1-MIDAS, and thus also for the Rea1-dependent pre-60S restructuring, are highly conserved acidic residues in the Rsa4-UBL (E114) or Ytm1-UBL (E80)[25,38], which mimic the crucial D or E interaction residues in extracellular ligands as part of the R-G-D peptide motif, which form strong noncovalent bonds with the MIDAS domain of the αI integrin subunit[31,35,36]. The interaction between human Ytm1 (WDR12) and human Rea1-MIDAS, which is dependent on coordination of a divalent metal ion and is sensitive to mutational perturbation of a critical residue in WDR12 (E78), forms also during the human ribosome biogenesis pathway[39].

The structure of the nucleoplasmic pre-60S substrate (the Rix1–Rea1-containing pre-ribosomal particle), from which the assembly factor Rsa4 is removed by Rea1, was resolved by cryo-EM[22,23]. Recently, cryo-EM structures of *Schizosaccharomyces pombe* (*Sp*) and *Saccharomyces cerevisiae* (*Sc*) Rea1 were reported, revealing different conformational stages of Rea1[28,29]. For *Sp*Rea1 it was shown that in presence of ATP and the chemical inhibitor ribozinoindole-1 (Rbin-1)[40], the C-terminal Rea1-MIDAS docks onto the N-terminal AAA$^+$ ring[28]. The docking of the substrate-binding MIDAS onto the AAA$^+$ ring was also found for *Sc*Rea1 in the apo and AMPPNP bound states upon deletion of an α-helical bundle extension (helix 2 insertion) that protrudes from the Rea1-AAA2 domain[29]. Moreover, it was suggested that the Rea1 ring and MIDAS domain are in direct contact on the Rix1–Rea1 pre-ribosome, which would allow nucleotide-state-dependent conformational changes in the AAA$^+$ ring to be directly transmitted to the MIDAS and its UBL-containing substrate proteins[28,29].

In this study, we report crystal structures of the *Chaetomium thermophilum* (*Ct*) Rea1-MIDAS, both alone and in complex with its pre-ribosomal substrates, the *Ct*Ytm1- or *Ct*Rsa4-UBL domains. Besides displaying the typical MIDAS fold known for integrins, the *Ct*Rea1-MIDAS crystal structures reveal evolutionary conserved Rea1 specific structural elements. One element forms a β-hairpin upon complex formation with the UBL domains of Rsa4 and Ytm1, thereby providing an anchor to hold the UBL domain in place. A second Rea1-MIDAS-specific element is a conserved loop with a nuclear localization signal (NLS) that promotes nuclear import of full-length Rea1. However, the MIDAS loop carries a second function, essential for the correct positioning of the MIDAS domain onto the Rea1 AAA$^+$ ring and for triggering Rsa4 release from the pre-ribosome. Altogether, these findings give structural and functional insights how Rea1 removes assembly factors from the pre-ribosome to trigger ribosome maturation.

## Results

**Crystal structure of thermophile Rea1-MIDAS domain**. To gain insight into the structural phases in which Rea1 ATPase removes the assembly factors Rsa4 and Ytm1 from pre-60S particles, we initially tried to crystallize the *S. cerevisiae* Rea1-MIDAS domain. However, due to the low yield and solubility of this construct, we switched to the thermostable Rea1-MIDAS from *C. thermophilum* (*Ct*) (Fig. 1a and Supplementary Fig. 1a). This construct (*Ct*Rea1-MIDAS) carries a ~105 residue-long N-terminal extension (Supplementary Fig. 1a), which is absent from integrin MIDAS domains but is known to contribute to Ytm1 and Rsa4 binding both in yeast and human (Supplementary Fig. 1b)[38,39]. We were able to purify the *Ct*Rea1-MIDAS construct in sufficient amount to allow us to obtain crystals and determine its crystal structure at 2.3 Å (Fig. 1b, for data collection and refinement statistics, see Table 1). The core structure of the *Ct*Rea1-MIDAS is similar to MIDAS domains of integrins (most similar to the A1 domain of von Willebrand factor, PDB ID: 1SQ0 [https://doi.org/10.2210/pdb1SQ0/pdb]; RMSD of 2.9 Å for 176 residues), exhibiting a typical α/β Rossmann fold with six β-strands and six α-helices (Fig. 1c). However, the *Ct*Rea1-MIDAS contains three additional conserved structural elements (termed elements I, II and III) that are not present in integrin-MIDAS domains (Fig. 1c and Supplementary Fig. 2). The first half of the Rea1-specific N-terminal extension contains a long α-helix (element I), which wraps around the MIDAS domain core. Although this part of the N-terminal extension (amino acids 4693–4732) has a low-sequence conservation, it contains an invariant tryptophan residue in the first part of the helix (W4710 in *C. thermophilum* and

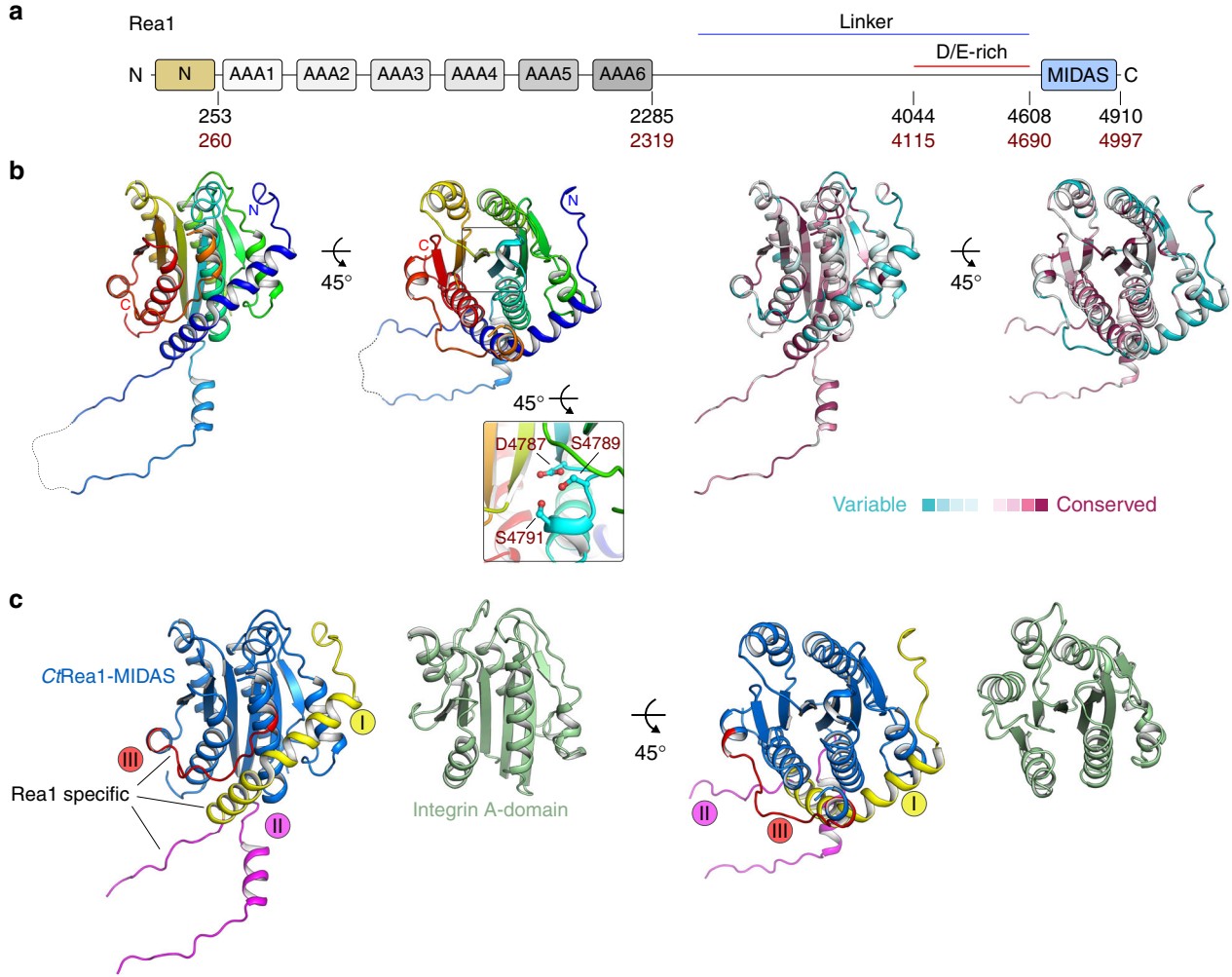

**Fig. 1** The crystal structure of the extended *Ct*Rea1-MIDAS reveals novel Rea1-specific elements. **a** Domain architecture of Rea1. The domain boundaries are indicated by the residue numbers below for *S. cerevisiae* (black) and *C. thermophilum* (red). **b** Crystal structure of the Rea1-MIDAS domain from *C. thermophilum* in two orientations. The close-up view highlights the DxSxS motif required for coordination of the divalent metal ion (left-hand structures). A ConSurf analysis of the *Ct*Rea1-MIDAS domain is shown in the two right-hand structures. Conserved amino acids, maroon; variable amino acids, turquoise. **c** Structure comparison of the *Ct*Rea1-MIDAS domain and the integrin A-domain (PDB ID: 1IDO) in two orientations. The integrin A-domain, green; the classical MIDAS fold of Rea1, blue; Rea1-specific elements I–III, yellow, purple, and red, respectively

W4633 in *S. cerevisiae*), which packs between helices α3 and α4 and most likely anchors the N-terminal helix in place (Supplementary Fig. 1a and Supplementary Fig. 3). Mutation of this tryptophan residue to arginine in *Sc*Rea1 revealed a crucial role, with impaired cell growth and loss of Rea1-MIDAS binding to Rsa4 (Supplementary Fig. 3). The second Rea1-specific MIDAS element (II, termed loop) forms a 45 amino-acid-long disordered loop (amino acids 4733–4777) opposite the critical DxSxS motif (Fig. 1b, c and Supplementary Fig. 1a). Notably, this loop is one of the most conserved regions within the Rea1-MIDAS domain, with 84% identity between yeast and human. The third MIDAS element (III) is located in the C-terminal region preceding the last α-helix of the classical α/β Rossmann fold (Fig. 1b, c and Supplementary Fig. 1a). This sequence shows a higher degree of variation between different Rea1 homologs and is predominantly disordered within the MIDAS apo structure.

**Rea1-MIDAS structures in complex with its ligands**. Next, we aimed to structurally investigate the complex between *Ct*Rea1-MIDAS and its two ligands, the ubiquitin-like domains (UBL) of *Ct*Rsa4 and *Ct*Ytm1. However, purification and crystallization proved to be challenging and might have been impeded by the

conserved but flexible MIDAS loop (element II). Since this loop (deletion of amino acids 4734–4773 in *Ct*Rea1, or 4657–4696 in *Sc*Rea1) is dispensable for MIDAS–Rsa4 interaction (Fig. 2a, b), we replaced it with a short Gly-Ser-Gly linker. Furthermore, we removed residues 4671–4689 from the *Ct*MIDAS N-terminus, which are not visible in our MIDAS apo-structure (Fig. 1b) and are not involved in the interaction with Rsa4. This Rea1-MIDAS Δloop construct indeed showed favorable biochemical properties with increased solubility, less degradation and a monodisperse behavior on gel filtration (Supplementary Fig. 4). For the co-crystallization trials, we also removed N-terminal residues from *Ct*Ytm1-UBL and *Ct*Rsa4-UBL that were not visible in previous crystal structures[26,37]. Using these trimmed constructs, we were able to reconstitute the complexes *Ct*Rea1-MIDASΔloop–*Ct*Rsa4-UBL and *Ct*Rea1-MIDASΔloop–*Ct*Ytm1-UBL, which readily crystallized, allowing us to solve their crystal structures at 1.89 and 1.84 Å, respectively (Fig. 2c, d and Supplementary Fig. 5). Although the overall structures of the Rea1-MIDAS Δloop domain in complex with either Rsa4-UBL or Ytm1-UBL are highly similar to the MIDAS apo structure, the terminal helices α1 and α9 were slightly tilted (as observed upon binding of integrins to its ligands for the C-terminal helix). The

**Table 1 Data collection and refinement statistics**

|  | *Ct*Rea1-MIDAS | *Ct*Rea1-MIDAS *Ct*Rsa4-Ubl | *Ct*Rea1-MIDAS *Ct*Ytm1-Ubl |
|---|---|---|---|
| *Data collection* |  |  |  |
| Beamline | ID30B | ID23-2 | ID23-2 |
| Wavelength (Å) | 0.97264 |  |  |
| Space group | $P4_12_12$ | $P4_32_12$ | $P2_1$ |
| *Cell dimensions* |  |  |  |
| *a, b, c* (Å) | 85.88, 85.88, 156.33 | 115.68, 115.68, 74.62 | 75.55, 83.17, 77.68 |
| $\alpha, \beta, \gamma$ (°) | 90, 90, 90 | 90, 90, 90 | 90, 117.72, 90 |
| Resolution (Å) | 44.54-2.33 (2.41-2.33) | 36.58-1.89 (1.95-1.89) | 39.62-1.84 (1.90-1.84) |
| $R_{merge}$ | 0.058 (1.158) | 0.144 (1.89) | 0.069 (0.879) |
| $R_{pim}$ | 0.032 (0.653) | 0.041 (0.522) | 0.031 (0.413) |
| $I/\sigma(I)$ | 14.09 (1.38) | 12.86 (1.49) | 13.33 (1.72) |
| CC1/2 | 0.999 (0.414) | 0.999 (0.478) | 0.999 (0.664) |
| Total reflections | 200499 (19385) | 543266 (56193) | 411974 (39036) |
| Unique reflections | 47603 (4730) | 41048 (4026) | 73123 (7287) |
| Completeness (%) | 99.79 (99.60) | 99.97 (100.0) | 99.02 (99.67) |
| Multiplicity | 4.2 (4.1) | 13.2 (14.0) | 5.6 (5.4) |
| *Refinement* |  |  |  |
| $R_{work}$ | 0.1815 (0.2912) | 0.1762 (0.3057) | 0.1879 (0.3095) |
| $R_{free}$ | 0.2121 (0.3388) | 0.2109 (0.3243) | 0.2103 (0.3432) |
| *No. of atoms* |  |  |  |
| Protein | 2291 | 2863 | 5368 |
| Solvent | 46 | 219 | 406 |
| Ligands | 23 | 58 | 20 |
| *B-factors* |  |  |  |
| Protein | 81.58 | 41.53 | 39.12 |
| Solvent | 87.54 | 50.51 | 45.10 |
| Ligands | 120.72 | 68.99 | 61.33 |
| *RMS deviations* |  |  |  |
| Bond lengths (Å) | 0.008 | 0.018 | 0.018 |
| Bond angles (°) | 1.00 | 1.42 | 1.50 |
| *Ramachandran statistics* |  |  |  |
| Most favored (%) | 97.86 | 98.01 | 96.98 |
| Disallowed (%) | 0 | 0.28 | 0 |

Values in parentheses refer to the highest resolution shell

$Mg^{2+}$-binding site in this case is fully ordered and the metal ion is coordinated cooperatively by the MIDAS and UBL domains (Fig. 2c, d). However, one unforeseen major structural re-arrangement concerns the Rea1-specific element III, which in the apo structure forms a loop that crawls along helices α1 and α9 and was not fully resolved, but in the ligand-bound form becomes re-arranged, the short helix α8 partially unfolds and forms a β-hairpin that provides an additional binding site for the UBL domains of Rsa4 and Ytm1 (Fig. 2c–e). This arrangement is only seen in the Rea1 MIDAS–ligand complexes and is not observed in integrin MIDAS–ligand structures.

To investigate the importance of this binding site for interaction with the ligand Rsa4, we mutated two residues (*Ct*Rea1 F4951 > A/R; *Ct*Rea1 I4959 > A/R), which lie within the β-hairpin and directly contact the Rsa4- or Ytm1-UBL (Fig. 3a, b). The interactions of these mutant *Ct*Rea1-MIDAS constructs with *Ct*Rsa4 were impaired, but mutation of other residues, V4956 > A/R or V4958 > A/R in Rea1, which are also part of the β-hairpin but not in direct contact with Rsa4, were still scored positive in a yeast two-hybrid assay (Fig. 3c). We also generated the equivalent β-hairpin mutations in the *S. cerevisiae* Rea1-MIDAS (*Sc*Rea1 Y4859 > A/R, I4871 > A/R). In the yeast two-hybrid analysis, these yeast Rea1 mutations also impaired the interaction with Rsa4 (Fig. 3d), and when tested in context of full-length Rea1, yeast cell growth was slowed (Fig. 3e). However, only the double mutants, in particular Rea1 Y4859R I4871R, but not the single amino acid substitutions, exhibited a significant growth defect, most evident at lower (16 and 23 °C) and higher (37 °C) temperatures (Fig. 3e).

To assess the functional relevance of the Rea1-MIDAS β-hairpin–UBL interactions, we performed genetic interaction studies with Rsa4 and Ytm1. The combination of the *rea1* Y4859 > A/R I4871 > A/R constructs with mutant alleles of either Rsa4 (*rsa4*-1) or Ytm1 (*ytm1* S78L)[25,38] yielded cells, which were no longer viable (Fig. 3f). These synthetic lethal growth defects suggest that the contact between the Rea1-MIDAS β-hairpin and the Rsa4- and Ytm1-UBL, as revealed in our crystal structures, plays an important role in pre-60S biogenesis in living cells.

Recently, a cryo-EM structure of the isolated full-length Rea1 molecule from *S. pombe* revealed that its C-terminal MIDAS domain was docked onto the hexameric AAA[+] ring in the presence of ATP and a Rea1-specific inhibitor Rbin-1[28]. This could be also seen in the context of the Rix1–Rea1 pre-60S particle, in which the Rea1-MIDAS domain that was modeled based on the integrin-MIDAS structure could be fit into an unassigned density on the Rea1-AAA[+] ring with direct contact to the Rsa4-UBL domain[28]. Based on our high-resolution X-ray structure, we could now fit very precisely the *Ct*Rea1-MIDAS–Rsa4-UBL structure into the cryo-EM density of the yeast Rix1–Rea1 pre-60S particle, which revealed that the Rea1-MIDAS β-hairpin directly contacts the Rsa4-UBL domain (Fig. 3g). This supports our suggestion that the β-hairpin in the Rea1-MIDAS domain provides an additional anchor to hold the Rsa4-UBL ligand in place.

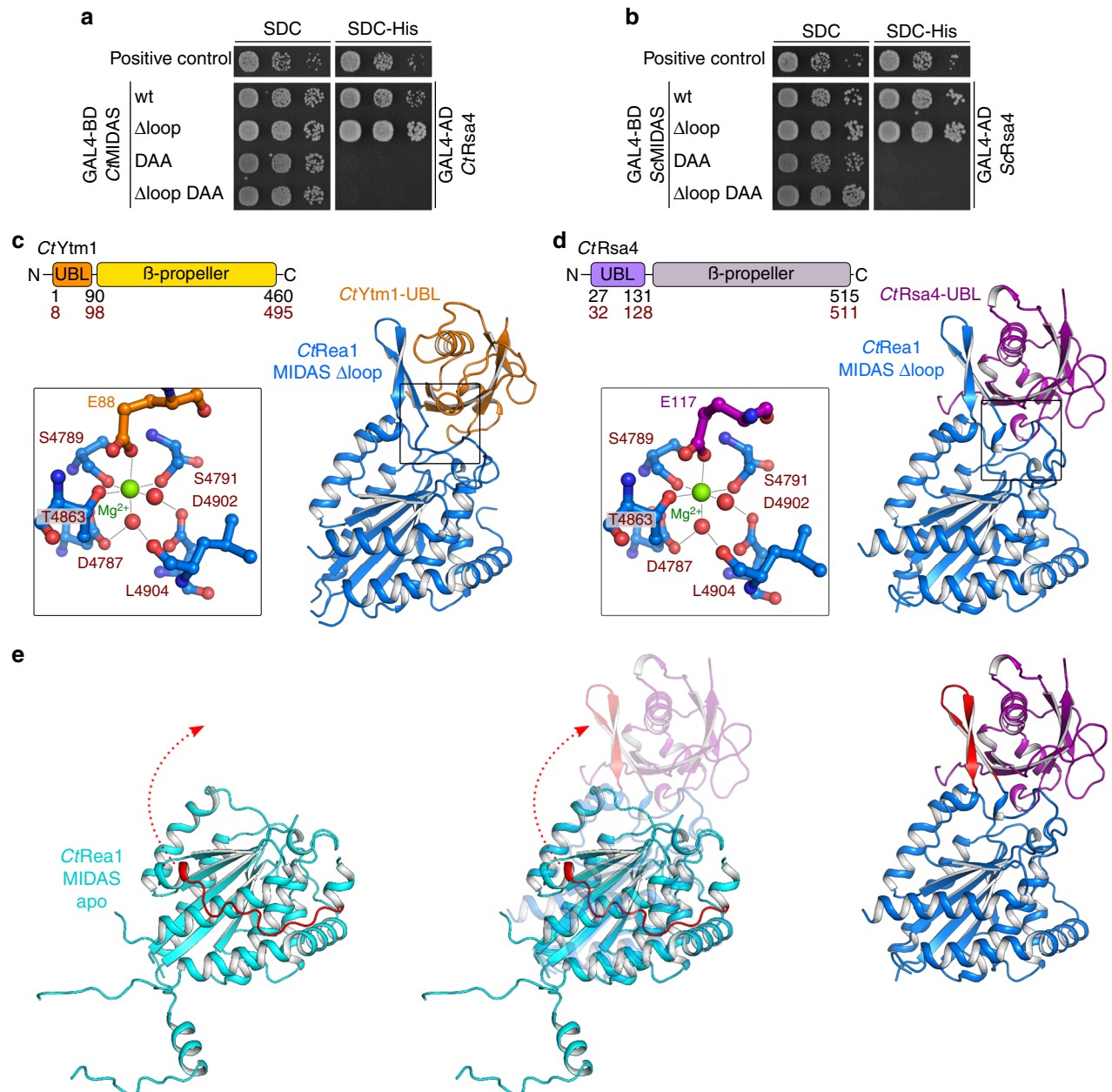

**Fig. 2** The crystal structure of the *Ct*Rea1-MIDAS Δloop with its ligands *Ct*Rsa4-UBL and *Ct*Ytm1-UBL. **a, b** Yeast two-hybrid analysis of the interactions between the indicated Rea1-MIDAS constructs and full-length Rsa4 from *C. thermophilum* (**a**) and *S. cerevisiae* (**b**). The Rea1-MIDAS constructs were fused to an N-terminal GAL4-BD (binding domain) and the Rsa4 constructs were fused to an N-terminal GAL4-AD (activation domain). Plasmids were co-transformed into the PJ69-4A yeast two-hybrid strain and representative transformants were spotted in tenfold serial dilutions on SDC (SDC-Leu-Trp) and SDC-His (SDC-Leu-Trp-His) plates. Cell growth was monitored after incubation for 3 days at 30 °C. Co-transformation of p53 (residues 72–390) fused to the GAL4-BD and SV40 (residues 84–708) fused to the GAL4-AD served as a positive control. **c, d** Crystal structures of the Rea1-MIDAS domain lacking the protruding element II loop in complex with the UBL domains of Ytm1 (**c**) and Rsa4 (**d**) from *C. thermophilum*. The domain organization of Ytm1 and Rsa4 are shown above the structures. The residue numbers indicate the domain boundaries for *S. cerevisiae* (black) and *C. thermophilum* (red). Zoomed-in views of the $Mg^{2+}$-coordinating residues are shown left of the X-ray structures. The $Mg^{2+}$ ion is shown in green, the amino acids of the MIDAS consensus motif in red and the glutamate within Ytm1 and Rsa4, which is essential for binding to the MIDAS, in orange (**c**) and purple (**d**), respectively. **e** Comparison of the Rea1-MIDAS structure and the Rea1-MIDAS Δloop structure in complex with the Rsa4-UBL. The Rea1-specific element III is shown in red, and the rearrangement from its disordered state in the MIDAS apo structure to a β-hairpin in the complex with the Rsa4-UBL is shown

**Rea1-MIDAS loop essential for pre-60S assembly.** Next, we addressed the in vivo role of the highly conserved Rea1-MIDAS loop (element II) that is dispensable in vitro for formation of complexes with Rsa4 or Ytm1. For this purpose, we deleted this loop from full-length *Sc*Rea1, and expressed it as a TAP-Flag-tagged construct in yeast, either under its own promoter or the regulatable *GAL1-10* promoter (Fig. 4a, b). Replacement of the loop by a short Gly-Ser-Gly linker (Rea1Δloop) caused a lethal phenotype in yeast, as did deletion of the entire MIDAS (Rea1ΔMIDAS, amino acids 1–4621) or mutation of the conserved $Mg^{2+}$-binding motif DxSxS to DxAxA (Rea1-DAA; Fig. 4a). However, overexpression of Rea1ΔMIDAS or Rea1Δloop did not cause a dominant-negative growth phenotype, in contrast to the either slight or strong dominant growth defects observed

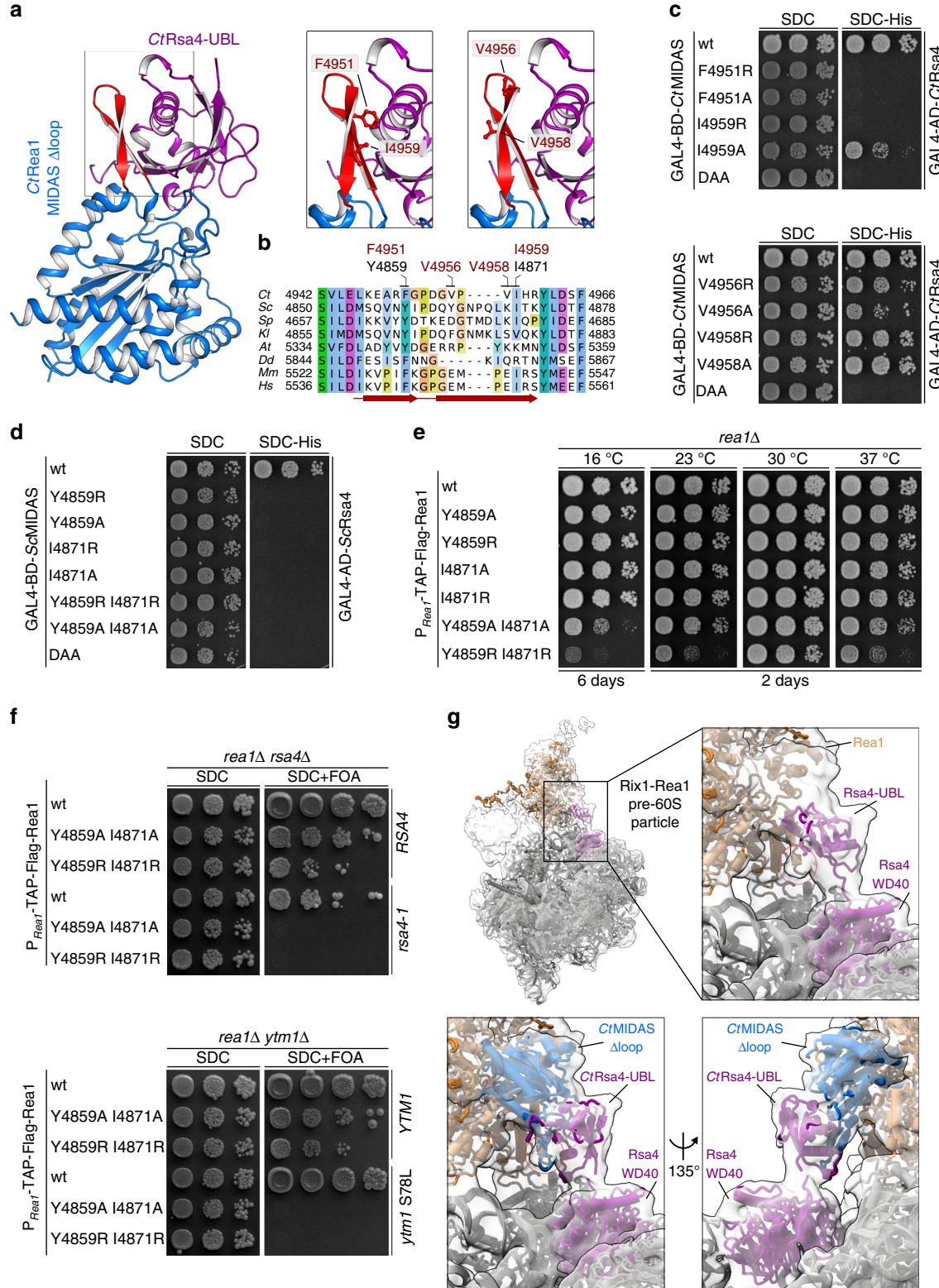

with the Rea1-DAA or Rea1-AAA3 Walker A (K1089A) mutants, respectively (Fig. 4b). Thus, since the Rea1-MIDAS loop is essential but its deletion in full-length Rea1 does not confer a dominant-negative phenotype, we conclude that this Rea1ΔLoop construct cannot compete with wild-type Rea1 for association with pre-60S particles. To prove this assumption, we affinity-purified Rea1ΔMIDAS and Rea1ΔLoop from yeast cells. This demonstrated that Rea1ΔMIDAS and Rea1ΔLoop mutants do not co-enrich on pre-60S particles (Fig. 4c).

## PY-NLS in the MIDAS loop mediating Rea1 nuclear import.

To discover why the lethal Rea1ΔLoop mutant does not associate with pre-60S particles, although in principle it should

**Fig. 3** The Rea1-MIDAS-specific element III forms a novel interaction site with the UBL domain. **a** Crystal structure of the *Ct*Rea1-MIDAS Δloop *Ct*Rsa4-UBL complex (left panel) and close-up of the interaction between the MIDAS element III and the Rsa4-UBL domain (right panels). The residues selected for mutational analysis are indicated. **b** Multiple sequence alignment of Rea1 showing the Rea1-MIDAS-specific element III. Sequences of *Chaetomium thermophilum* (*Ct*), *Saccharomyces cerevisiae* (*Sc*), *Schizosaccharomyces pombe* (*Sp*), *Kluyveromyces lactis* (*Kl*), *Arabidopsis thaliana* (*At*), *Dictyostelium discoideum* (*Dd*), *Mus musculus* (*Mm*), and *Homo sapiens* (*Hs*) were aligned with Clustal Omega and visualized with Jalview. The element III with its two β-sheets are indicated in red, and residues mutated in this study are shown above the alignment (red: *C. thermophilum*, black: *S. cerevisiae*). **c**, **d** Yeast two-hybrid analysis between the indicated Rea1-MIDAS constructs and full-length Rsa4 from *C. thermophilum* (**c**) and *S. cerevisiae* (**d**). The MIDAS constructs were fused to an N-terminal GAL4-BD and the Rsa4 constructs to an N-terminal GAL4-AD. The constructs were co-transformed into yeast (PJ69-4A) and cells were spotted on SDC (SDC-Leu-Trp) and SDC-His (SDC-Leu-Trp-His) plates. Cell growth was inspected after 3 days at 30 °C. **e** The indicated Rea1 constructs were transformed in a *rea1Δ* shuffle strain. After plasmid shuffling on SDC plates containing 5-FOA, cells were spotted on YPD plates and growth was monitored at the indicated temperatures and times. **f** The *rea1Δ rsa4Δ* and *rea1Δ ytm1Δ* double shuffle strains were transformed with the indicated wild-type and mutant constructs. Transformants were spotted on SDC (SDC-Leu-Trp) and SDC + FOA plates and cell growth at 30 °C was monitored after 3 and 5 days, respectively. **g** Electron density map of the Rix1–Rea1 particle (EMD-3199) and the models of Rsa4 and Rea1 (PDB ID: 5JCS) in purple and orange, respectively. A density not occupied by the model is highlighted with a red circle (upper close-up). The lower close-up views show the rigid body fit of the Rea1-MIDAS Δloop–Rsa4-UBL complex from *C. thermophilum* within the electron density map of the Rix1–Rea1 particle (EMD-3199). The β-hairpin formed by the Rea1-specific element III is highlighted with a red circle

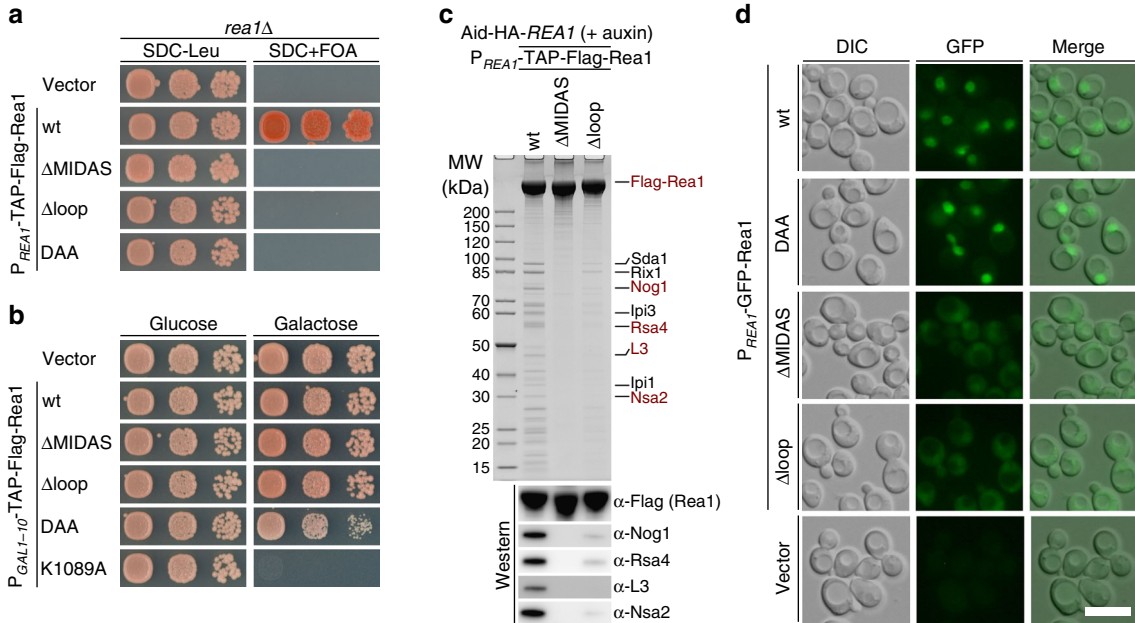

**Fig. 4** The conserved Rea1-MIDAS element II loop is required for Rea1's essential function. **a** The indicated constructs were transformed in a *rea1Δ* shuffle strain and the viability of the respective mutants assessed on 5-FOA-containing plates. Cells were grown at 30 °C for 3 (SDC-Leu) and 5 days (SDC + FOA). **b** The indicated constructs under control of the *GAL1-10* promoter were transformed in a wild-type strain (expressing endogenous *REA1*) and cell growth was monitored after incubation at 30 °C on plates containing glucose (SDC-Leu, left) or galactose (SGC-Leu, right) for 2 and 3 days, respectively. **c** Affinity purification of the indicated Rea1 constructs fused to an N-terminal TAP-Flag tag and under control of the endogenous *REA1* promoter. The constructs were transformed into an AID-HA-*REA1* degron strain. The endogenous Rea1 was depleted by the addition of auxin (final concentration 500 μM) to the cultures 2 h before cells were harvested. Final eluates were analyzed by SDS-PAGE and Coomassie staining or by western blot analysis against pre-60S assembly factors and ribosomal protein L3, using the indicated antibodies. **d** The subcellular localization of the indicated full-length Rea1 constructs N-terminally fused to GFP was monitored by fluorescence microscopy. Scale bar: 5 μm. Source data are provided as a Source Data file

be able to (see above), we tested whether nuclear import of Rea1Δloop is inhibited. We considered the possibility that Rea1Δloop is not imported into the nucleus, because the deleted MIDAS loop contains a predicted PY-NLS with the typical signature pattern ($R/K/H-X_{2–5}-PY/L$), which we notice in all Rea1 orthologues from yeast to human (Fig. 5a). Consistent with this bioinformatics-based expectation, green fluorescent protein (GFP)-tagged versions of wild-type or Rea1-DAA exhibited a pronounced nuclear localization, whereas GFP-labeled Rea1ΔMIDAS and Rea1Δloop constructs mislocalized to the cytoplasm (Fig. 4d). Moreover, the Rea1-MIDAS loop sequence from both *S. cerevisiae* and *C. thermophilum*, when fused to 3×GFP, induced robust nuclear localization (Fig. 5b).

Subsequently, we performed in vitro binding assays with GST-tagged *Ct*Kap104, which is the general nuclear import receptor (karyopherin/importin) of PY-NLS-carrying nuclear proteins[41,42]. The *Ct*Rea1-MIDAS was found to efficiently bind to *Ct*Kap104 (Fig. 5c, lane 6), similar to what has been observed for another PY-NLS-carrying protein, Syo1[43]; however, deletion of the Rea1-MIDAS loop impaired the interaction (Fig. 5c, lane 7). When the consensus PY motif (*Ct*Rea1 P4757, Y4758) or the critical consensus upstream lysine residues (*Ct*Rea1 K4753, K4754) in the *Ct*Rea1-MIDAS were mutated to alanine, impaired binding to *Ct*Kap104 was observed (Fig. 5d, lanes 9–11). Consistent with the biochemical data, the intracellular location of the Rea1-MIDAS loop-3×GFP reporter constructs carrying the PY-NLS mutations was altered, with *Ct*MIDAS KK/PY>A and

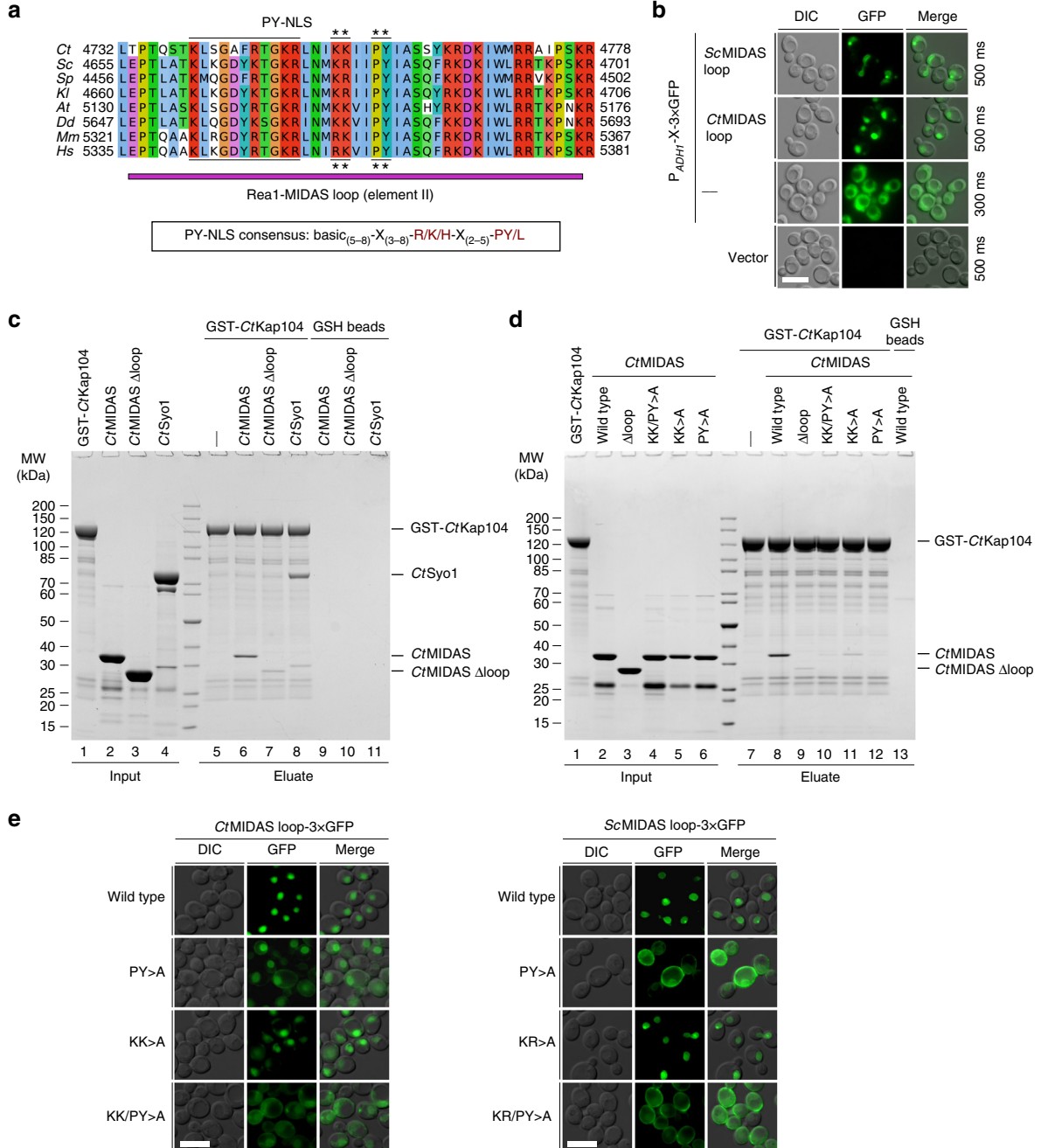

**Fig. 5** The Rea1 MIDAS element II loop harbors a conserved PY-NLS that interacts with Kap104. **a** Multiple sequence alignment of Rea1 showing the MIDAS loop region (purple bar). The PY-NLS is highlighted and the mutated residues are marked with an asterisk. The PY-NLS consensus sequence is shown below the alignment. The sequences of *Chaetomium thermophilum* (*Ct*), *Saccharomyces cerevisiae* (*Sc*), *Schizosaccharomyces pombe* (*Sp*), *Kluyveromyces lactis* (*Kl*), *Arabidopsis thaliana* (*At*), *Dictyostelium discoideum* (*Dd*), *Mus musculus* (*Mm*), and *Homo sapiens* (*Hs*) were aligned with Clustal Omega and visualized with Jalview. **b** The *Sc*MIDAS loop (residues 4655–4701) or the *Ct*MIDAS loop (residues 4732–4778) were fused to a 3×GFP reporter and the subcellular localization of fusion proteins was monitored by fluorescence microscopy. Nomarski (DIC), GFP and merged pictures are shown. Scale bar: 5 μm. **c**, **d** Binding assay between GST-*Ct*Kap104 and *Ct*Rea1-MIDAS, *Ct*MIDAS Δloop, the indicated *Ct*MIDAS PY-NLS point mutants, and *Ct*Syo1. After incubation with GSH-agarose, GST-*Ct*Kap104 and bound proteins were eluted with GSH and analyzed by SDS-PAGE and Coomassie staining. **e** The subcellular localization of the *Ct*MIDAS loop (left), the *Sc*MIDAS loop (right), or indicated PY-NLS point mutants fused to 3×GFP was monitored by fluorescence microscopy. Nomarski (DIC), GFP and merged pictures are shown. Scale bars: 5 μm. Source data are provided as a Source Data file

*Sc*MIDAS KR/PY>A (all four PY-NLS consensus residues mutated) showing the most striking cytoplasmic mislocalization (Fig. 5e). We conclude from these studies that the Rea1-MIDAS loop harbors a functional NLS that mediates nuclear import of the Rea1 AAA$^+$ ATPase, which is a prerequisite for assembly of Rea1 into pre-60S particles.

**Rea1-MIDAS loop triggers Rsa4 release from pre-60S particles**. Based on these findings, we examined whether it is possible to rescue the deletion of the PY-NLS from the Rea1-MIDAS loop by inserting another PY-NLS from an unrelated nuclear protein. Hence, we considered the well-studied PY-NLSs present in the yeast nuclear proteins L4 (also known as uL4), Syo1, and Hrp1

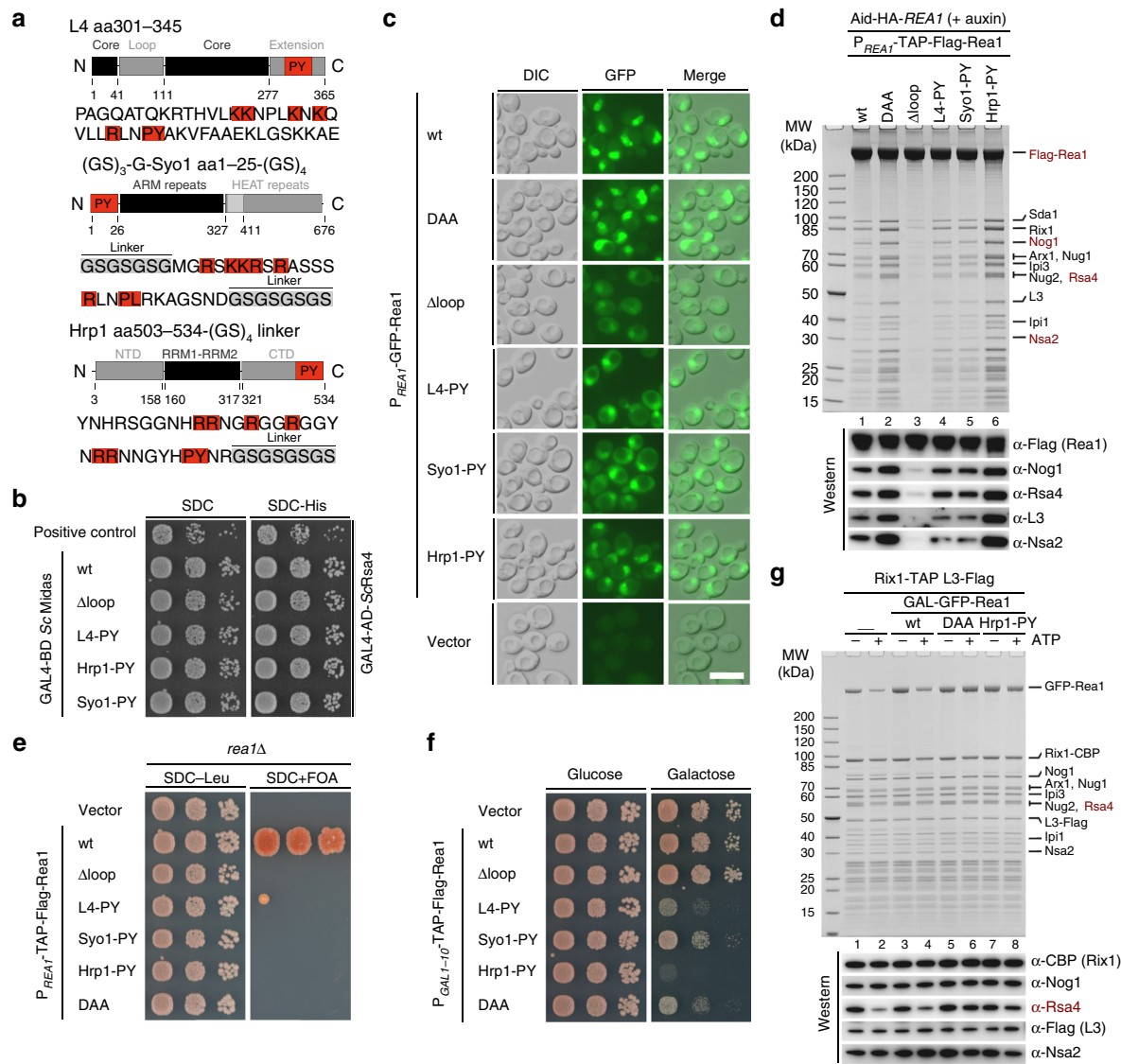

**Fig. 6** Nuclear import of Rea1 can be restored by a heterologous PY-NLS but still lacks Rea1-MIDAS loop-specific functions. **a** The PY-NLS containing Rea1-MIDAS element II loop (residues 4657–4696) was exchanged with PY-NLS-containing regions derived from either L4, Syo1, or Hrp1, whose domain organizations are shown. The different PY-NLS-containing fragments used to insert into the Rea1-MIDAS Δloop construct are highlighted in red and their amino acid sequences are shown below. Additional linker sequences are labeled and marked in gray. **b** Yeast two-hybrid analysis of the interactions between Rsa4 and the indicated Rea1-MIDAS constructs from *S. cerevisiae*. **c** Subcellular localization of the indicated N-terminally GFP-tagged Rea1 constructs under control of the endogenous *REA1* promotor was monitored by fluorescence microscopy. Nomarski (DIC), GFP and merged images are shown. Scale bar: 5 μm. **d** Affinity purification of the Rea1-MIDAS Δloop constructs containing foreign PY-NLSs from L4, Syo1, or Hrp1. The indicated Rea1 constructs were fused to an N-terminal TAP-Flag tag under the endogenous *REA1* promotor, which were transformed into an AID-HA-*REA1* degron strain. Rea1 depletion was induced by addition of auxin (500 μM final concentration) for 2 h before harvesting and subsequent affinity purification. Final eluates were analyzed by SDS-PAGE following Coomassie staining or by western blot analysis using the indicated antibodies. **e** The indicated Rea1 constructs were transformed in a *rea1Δ* shuffle strain and the viability of the respective mutants assessed on 5-FOA-containing plates. **f** The indicated Rea1 constructs under control of the *GAL1-10* promoter or an empty vector control were transformed in a wild-type strain (expressing endogenous *REA1*) and cell growth was monitored on plates containing glucose (SDC-Leu, left) or galactose (SGC-Leu, right). **g** Rix1-TAP pre-60S particles isolated after overexpression of the indicated Rea1 constructs under control of the *GAL1-10* promotor (lanes 3–8) were incubated with 2.5 mM ATP (lanes 2, 4, 6, 8) or left untreated (lanes 1, 3, 5, 7) before re-isolation via L3-Flag. Final Flag-eluates were analyzed by Coomassie staining or by western blotting using the indicated antibodies. Source data are provided as a Source Data file

(Fig. 6a)[43–45]. Insertion of either of these heterologous PY-NLSs into the Rea1Δloop construct, which did not impair binding to Rsa4 (Fig. 6b), not only restored nuclear import of Rea1 into the nucleus (Fig. 6c), but also allowed assembly of Rea1 into pre-60S particles (Fig. 6d). To our surprise, however, these Rea1 constructs were not able to complement the otherwise lethal *rea1Δ* null yeast strain (Fig. 6e), suggesting an additional and, most importantly, essential role for the Rea1-MIDAS loop. In support

of this idea, overexpression of Rea1 carrying the heterologous PY-NLSs induced dominant-negative growth defects (Fig. 6f), which might mean that the altered loop sequences cannot confer the Rea1-loop-specific function on the pre-60S particle.

This logic prompted us to ask whether release of Rsa4 from the pre-60S particle requires the essential Rea1 loop sequence, even though it is not required for the high-affinity Rea1-MIDAS–Rsa4-UBL interaction (Fig. 2a, b). To analyze this directly, we

affinity-purified pre-60S particles containing either wild-type Rea1, Rea1-DAA, or Rea1Δloop with re-implanted Hrp1 PY-NLS and performed an in vitro assay that measures ATP-dependent removal of Rsa4. Strikingly, we did not observe a significant ATP-dependent release of Rsa4 from pre-60S particles, which carry the Rea1 AAA$^+$ ATPase with the Hrp1 PY-NLS-modified MIDAS loop (Fig. 6g, lane 8). This situation is similar to the Rea1 DAA mutant, which is not able to bind and hence cannot extract the Rsa4-UBL in the ATP-dependent reaction in vitro (Fig. 6g, lane 6).

**Rea1-MIDAS loop interacts with the Rea1 AAA$^+$ ring.** To further dissect the role of the Rea1-MIDAS loop, we sought to reconstitute the interaction between the Rea1-AAA$^+$ ring, the MIDAS domain and either Rsa4-UBL or Ytm1-UBL, based on purified constructs (Fig. 7a, input lanes 1–4 and Supplementary Fig. 6a, input lanes 1–5), and test whether the MIDAS loop contributes to this interaction. In fact, it was possible to assemble trimeric *Ct*Rea1-MIDAS–*Ct*Rsa4-UBL–*Ct*Rea1-NAAA$^+$ or *Ct*Rea1-MIDAS–*Ct*Ytm1-UBL-*Ct*Rea1-NAAA$^+$ complexes in vitro (Fig. 7a, lane 7 and Supplementary Fig. 6a), although the Rea1-NAAA$^+$ ring did not associate stoichiometrically, even in the presence of ATP, AMPPNP or ATP and an inhibitor (Rbin-1) known to bind to the *Sp*Rea1-AAA$^+$ ring[40] (Fig. 7a, lanes 8 and 9 and Supplementary Fig. 6b), suggesting that additional factors or the pre-60S surface structure might play a role in strengthening this interaction. However, the MIDAS loop is clearly required to establish the association between the Rea1-NAAA$^+$ ring and the Rea1-MIDAS–Rsa4-UBL or Rea1-MIDAS–Ytm1-UBL heterodimer, because upon loop deletion only background binding was observed (Fig. 7a, compare lanes 7–9 with 10–12; Supplementary Fig. 6a), which might correspond to a low-affinity interaction between the Rsa4-UBL and the Rea1-NAAA$^+$ ring. Altogether, these data point to a direct role of the MIDAS loop for docking the entire Rea1-MIDAS domain carrying its Rsa4-UBL ligand to the Rea1-AAA$^+$ ring (Fig. 7d).

Encouraged by these findings, we attempted to identify a distinct electron density in the recently published *S. pombe* Rea1 cryo-EM structure that corresponds to the Rea1-MIDAS loop. Rigid body fitting of the *Ct*Rea1-MIDAS into the density map of *S. pombe* Rea1 unambiguously identified the Rea1-MIDAS loop attached to the Rea1-AAA$^+$ inner pore ring (Fig. 7b, c). Moreover, we suggest that a previously assigned α-helix of the Rea1-MIDAS model in the *S. pombe* Rea1 ring, which was created based on an integrin-MIDAS template[28], corresponds to the MIDAS β-hairpin (element III), which we clearly observe in the *S. pombe* Rea1 density map (Fig. 7b, right panels). Notably, the formation of this β-hairpin in our crystal structure required a complex to form between *Ct*MIDAS and *Ct*Rsa4-UBL, in contrast to its unstructured conformation in the *Ct*MIDAS apo-structure. This result suggests that the MIDAS domain, when attached to the Rea1-AAA$^+$ ring, is already in a primed stage allowing stable recruitment of either of Rea1's UBL ligands, Rsa4, or Ytm1.

## Discussion

This study revealed molecular insights into how the Rea1 AAA$^+$ ATPase interacts with its substrate proteins, Rsa4 and Ytm1, which upon a mechanochemical cycle dependent on ATP hydrolysis release these two assembly factors from the maturing pre-60S particles (Fig. 7d). We provide the atomic structures of the essential Rea1-MIDAS domain alone and in complex with its binding partners, the UBLs of Rsa4 and Ytm1. Moreover, we identified Rea1-specific elements in the conserved C-terminal MIDAS domain, which do not exist in the structurally related MIDAS domains from integrins, and investigated their functional role.

Compared to integrin MIDAS structures, the Rea1-MIDAS contains unique conserved structural elements. One such specific element is a β-hairpin (element III) that forms upon interaction with the Rsa4-UBL or Ytm1-UBL, whereas it is unstructured in the Rea1-MIDAS apo structure. This element is required for optimal cell growth in yeast and it contributes to the high-affinity of the MIDAS–UBL interaction. The second Rea1-specific MIDAS feature is an extended loop protruding from the MIDAS domain, which harbors a previously unrecognized PY-NLS for driving Rea1 into the nucleus. However, this NLS is highly conserved, which is normally not observed for other PY-NLSs. Our experimental analyses revealed a molecular explanation for this, pointing to a second specific function, which is essential for the mechanochemical cycle of the Rea1 AAA$^+$ ATPase. Specifically, the protruding MIDAS loop acts to tether the entire MIDAS domain to the Rea1 AAA$^+$ ring. Recent cryo-EM structures of *S. pombe* and *S. cerevisiae* Rea1 revealed that the Rea1 MIDAS can bind to the AAA$^+$ domain in a closed ring conformation[28,29]. Interestingly, the α-helical insert within helix 2 (H2) of the AAA2 domain interacts with the pore of the AAA$^+$ ring in the AMPPNP or ADP state within these structures, thereby trapping the Rea1 molecule in an open ring conformation, which, as a consequence, impairs the interaction of the MIDAS with the AAA$^+$ ring. The AAA2 H2 insert was originally described by us as an interaction site for the Rix1 subcomplex, and hence could be involved in the recruitment of Rea1 to pre-60S particles[23]. Upon binding of Rea1 to the pre-60S particle, the AAA2 H2 insert might be displaced from the AAA$^+$ ring, perhaps facilitated by binding to the Rix1 C-terminus, which in addition could facilitate AAA$^+$ ring closure and docking of the MIDAS domain onto the AAA$^+$ ring. However, we did not observe an increased binding of a *Ct*Rea1 AAA$^+$ mutant lacking the H2 insertion in AAA2 to the *Ct*Rea1-MIDAS–Rsa4-UBL heterodimer in vitro (V.M. and E.H., unpublished results), leaving it open whether this is the exact mechanism in vivo, or whether additional requirements are needed, which we cannot yet simulate by in vitro reconstitution. Similarly, the precise role of the MIDAS loop for Rea1 activity and Rsa4 removal remains unclear. It is tempting to speculate that ATP hydrolysis in the AAA$^+$ ATPase ring creates a pulling force on the MIDAS loop, which transmits mechanochemical energy to the attached UBL domain of Rsa4, thereby dislodging Rsa4 from the pre-60S particle. Since Rea1 is bound to the pre-60S subunit through several additional contacts (Rix1, L11, rRNA helix 38), all these interactions could be involved in the regulation of its nucleotide loading and enzymatic activity state by either blocking or promoting its energy-dependent function.

For future studies, our structural data might aid the discovery of inhibitors that impair the Rea1-MIDAS–Rsa4 or Rea1-MIDAS–Ytm1 interactions. Such inhibitors might be useful for further clarifying the mechanism of the Rea1 mechanochemical cycle, and also potentially used in the human system to target cells in disease stages including cancer.

## Methods

**Plasmids and strains.** Plasmids were constructed using standard recombinant DNA techniques. All plasmids used and constructed in this study are listed in Supplementary Tables 1 and 2. The *S. cerevisiae* strains used in this study were generated by standard tagging and gene disruption methods and are listed in Supplementary Table 3.

**Protein expression and purification from *E. coli*.** Proteins were expressed in *E. coli* Rosetta 2 (DE3) cells in 2YT medium supplemented with chloramphenicol (34 μg/ml) and either kanamycin (30 μg/ml) or ampicillin (100 μg/ml). Cells were grown to an $OD_{600}$ value of 0.8–1.0 at 37 °C, then maintained at 18 °C. Expression was induced with the addition of 0.4 mM IPTG, and cells were grown further overnight, harvested by centrifugation, and the cell pellets either used immediately

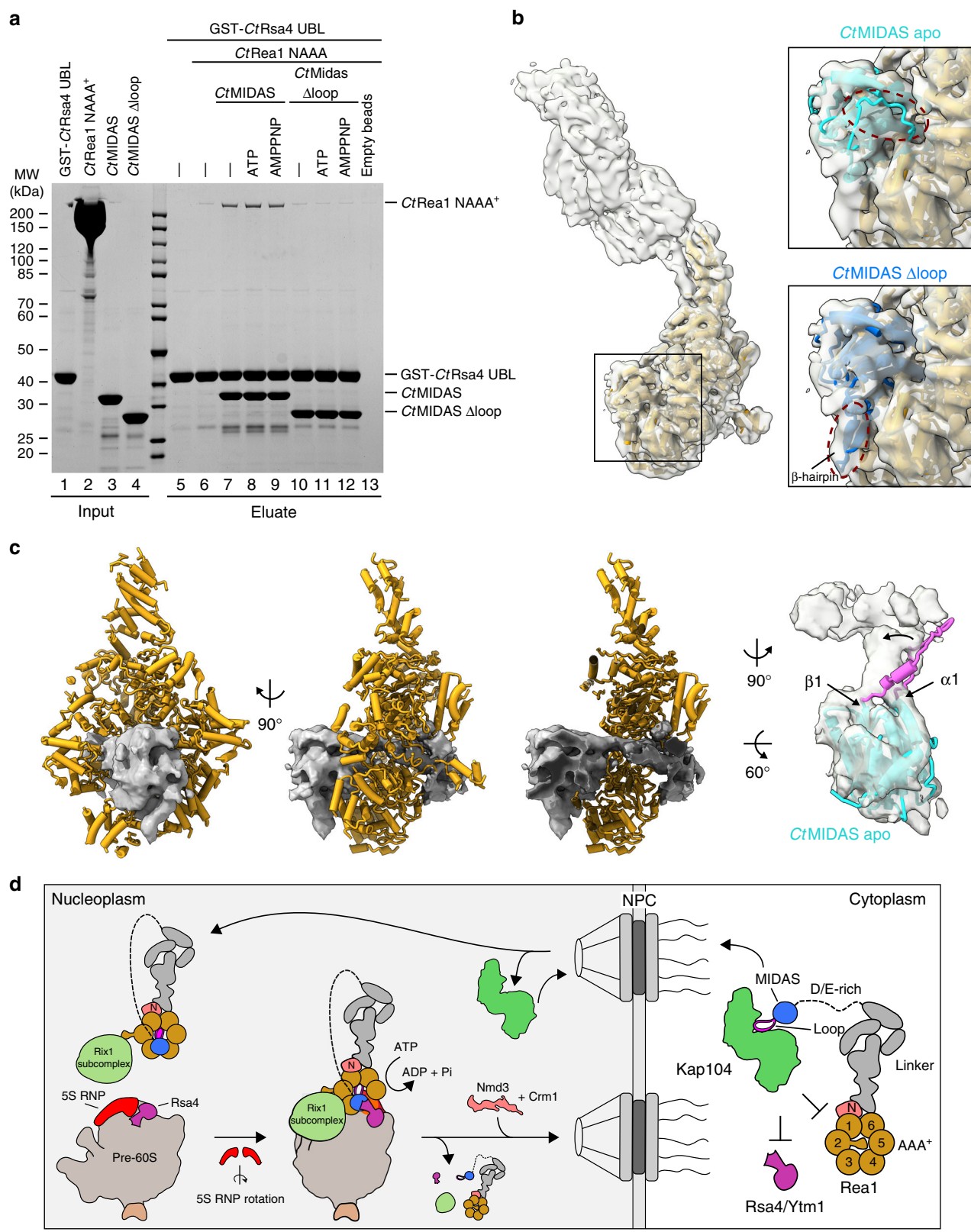

for lysis and purification or frozen with liquid nitrogen and stored at −20 °C. All variants of *Ct*Rea1-MIDAS were purified via a N- or C-terminal His₆ tag using metal-affinity chromatography (on Ni–NTA) and size-exclusion chromatography (SEC). In brief, cells were resuspended in lysis buffer (30 mM HEPES, 30 mM imidazole, 500 mM NaCl) and lysed with a Microfluidizer homogenizer (Microfluidics) at 0.55 MPa. The lysate was centrifuged for 35 min at 35,000 × *g* and 4 °C. The supernatant was applied to a 2 ml Ni–NTA column, washed with 50 column volumes of lysis buffer, and eluted with elution buffer (lysis buffer containing

400 mM imidazole). The Ni–NTA eluate was applied to a Superdex 200 26/60 column equilibrated with SEC buffer I (10 mM HEPES, pH 7.5, containing 500 mM NaCl). Peak fractions containing *Ct*Rea1-MIDAS were pooled, concentrated to 10–15 mg/ml and either used directly or frozen with liquid nitrogen and stored at −80 °C. The UBL domains of *Ct*Ytm1 and *Ct*Rsa4 were purified similarly to *Ct*Rea1-MIDAS variants, with the following changes: SEC was performed with a Superdex 75 26/60 column equilibrated with SEC buffer II (10 mM HEPES, pH 7.5, containing 150 mM NaCl); the complex formed between *Ct*Rea1-MIDAS and

**Fig. 7** The MIDAS element II loop is required for interaction of the Rea1-MIDAS domain with the Rea1-AAA[+] ring. **a** GST-tagged *Ct*Rsa4-UBL was incubated with *Ct*Rea1-MIDAS or MIDASΔloop and the *Ct*Rea1-NAAA[+] domain. Indicated nucleotides were added to a final concentration of 2 mM. The resulting complexes were isolated by incubation with GSH-agarose. GST-*Ct*Rsa4 and bound material was eluted with GSH and analyzed with SDS-PAGE and Coomassie staining. Protein inputs are shown in lanes 1–4, final eluates in lanes 5–13. **b** Left: cryo-EM density and model of the AAA[+] ATPase ring of *Sp*Rea1 (Mdn1) in presence of ATP and Rbin-1 (PDB ID: 6EES, EMD-9036). Right: close-up views highlighting the rigid-body fits of the *Ct*MIDAS apo (turquoise) and the *Ct*MIDASΔloop (blue) structures into the *Sp*Rea1 density. The model of the *Sp*Rea1-MIDAS was omitted for clarity. No density was observed for the disordered loop of the Rea1-specific element III within the MIDAS apo structure. Instead, the formed β-hairpin within the MIDAS Δloop structure fits into a density underneath the MIDAS domain. **c** Model of *Sp*Rea1 (PDB ID: 6EES), lacking the MIDAS domain, including the cryo-EM density (EMD-9036) of the MIDAS and contacting unaccounted-for density within the AAA[+] pore in two orientations and as cutaway view (left three panels). The right view shows the rigid body fit of the *Ct*MIDAS apo structure within the density. The unassigned density is directly connected to the densities of α1 and β1 of the MIDAS domain, strongly suggesting the loop (purple) of the Rea1-specific element II binds in the pore of the AAA[+] ring. **d** Model of Rea1 import and assembly into pre-60S particles. In the cytoplasm, the PY-NLS within the loop of the MIDAS domain attached to the unstructured D/E-rich domain is recognized by the importin Kap104, which delivers Rea1 into the nucleus. Upon Kap104 dissociation, the accessible MIDAS loop tethers the MIDAS onto the AAA[+] ring protruding into the ring pore, which together with formation of the β-hairpin-anchor primes the molecule for Rsa4-UBL interaction and release. At this stage, Rea1 is assembled to the nucleoplasmic pre-60S particle. Source data are provided as a Source Data file

*Ct*Ytm1-UBL was reconstituted with purified components (*Ct*Rea1-MIDAS/UBL molar ratio 1:7) in SEC buffer III (10 mM HEPES, 150 mM NaCl, 5 mM MgCl$_2$, pH 7.5), and subsequent removal of excess UBL by SEC using a Superdex 75 26/60 column equilibrated with SEC buffer III. Fractions containing the target complex were pooled, concentrated and used for crystallization trials.

**Crystallization and structure determination**. All crystals were grown with the sitting-drop vapor diffusion method at 18 °C. Prior to data collection, crystals were harvested in reservoir solution supplemented with 20% glycerol and flash-cooled with liquid nitrogen. Diffraction data were collected at various ESRF beamlines as detailed below. Data integration and reduction was performed with XDS[46] and AIMLESS[47]. Structure determination is outlined below for each structure.

Crystals of *Ct*Rea1-MIDAS variants were obtained from various solutions containing (NH$_4$)$_2$SO$_4$ (see below). Initial crystals diffracted to 4–5 Å and were twinned (apparent space group *P*321) and were difficult to reproduce. During our purification trials we noticed that the use of buffers with high salt concentrations resulted in less aggregation and increased the solubility of the protein; these trials were more reproducible and readily yielded crystals. After further optimization trials, well-diffracting crystals were obtained from a solution containing 1.48 M (NH$_4$)$_2$SO$_4$, 5 mM MgCl$_2$ and 10 mM NaI. Data were collected at the ESRF beamline ID30B[48] at a wavelength of 0.97264 Å. Crystals belong to the space group *P*4$_1$2$_1$2 with cell constants $a = b = 85.88$ Å, $c = 156.33$ Å, and $\alpha = \beta = \gamma = 90°$. Despite the rather short wavelength used for data acquisition, a significant anomalous signal could be detected and the structure could be solved de novo by iodide single-wavelength anomalous dispersion using SHELXC/D/E[49] navigated with HKL2MAP[50]. A clear solution (CC$_{all}$/CC$_{weak}$ of 30.00/17.40) was found with SHELXD. Phasing and density modification with SHELXE led to an interpretable electron density map that was used for automated model building with Buccaneer[51]. The majority of residues could be placed automatically and remaining residues were built manually with Coot[52]. The final model contains one molecule in the asymmetric unit. Three iodine sites were located, with occupancies of 1.0, 0.69, and 0.29. The first two sites are located along the (unformed) β-hairpin, while the third site is between β3 and α5.

Crystals of *Ct*Rea1-MIDASΔloop–*Ct*Rsa4-Ubl_32–128 grew within 1–2 days in a solution containing 2 M (NH$_4$)$_2$SO$_4$, 5% PEG 400 and 0.1 M MES at pH 6.5 with a protein concentration of 5.1 mg/ml. Diffraction data were collected at ESRF beamline ID23-2 (ESRF, Grenoble, France)[53]. Crystals belong to the space group *P*43212 with cell constants $a = b = 115.68$ Å, $c = 74.62$ Å and $\alpha = \beta = \gamma = 90°$. The structure was solved by molecular replacement as implemented in MOLREP[54] by first placing a truncated model of the *Ct*Rea1-MIDAS domain and then placing the *Ct*Rsa4-UBL domain (PDB ID: 4WJS[26] [https://doi.org/10.2210/pdb4WJS/pdb]). The final model contains one *Ct*Rea1-MIDASΔloop–*Ct*Rsa4-UBL complex.

Crystals of *Ct*Rea1-MIDASΔloop–*Ct*Ytm1-Ubl_8–98 grew after 20–30 days in a medium containing 3 M NaCl and 0.1 M Tris (pH 8.5) with a protein concentration of 16 mg/ml. Diffraction data were collected at ESRF beamline ID23-2. The structure was also solved by molecular replacement as implemented in MOLREP. Cell content analysis suggested the presence of two *Ct*Rea1-MIDASΔloop–*Ct*Ytm1-UBL complexes. Therefore initially, two molecules of *Ct*Rea1-MIDAS were placed and then two *Ct*Ytm1-UBL (PDB ID: 5EM2[37] [https://doi.org/10.2210/pdb5EM2/pdb]) domains were placed. The final model contains two *Ct*Rea1-MIDASΔloop–*Ct*Ytm1-UBL complexes. Refinement of all structures was performed with REFMAC5[55] and PHENIX[56]. Data collection and refinement statistics are summarized in Table 1. Structural comparisons were performed with GESAMT[57] from the CCP4[58] package. Interface analysis was done with PISA. Surface charge calculations were performed with the APBS[59] plugin within PyMOL. All structural figures were prepared with PyMOL or ChimeraX.

**Purification of the *Ct*Rea1 NAAA[+] ring**. The *Ct*Rea1 NAAA[+] ring construct (residues 1–2390) N-terminally fused to a protein A tag was overexpressed in yeast under the control of an *ADH1* promoter. Cells were grown in synthetic dextrose complete medium lacking leucine (SDC-Leu) to an OD$_{600}$ value of around 1.5. The medium was supplemented with 1.2% glucose, 1.2% bacto peptone, and 0.6% yeast extract (final concentrations) and grown to an OD$_{600}$ value of ~6. Cells were harvested and flash-frozen in liquid nitrogen. Cells were lysed in lysis buffer containing 30 mM HEPES (pH 7.5), 200 mM NaCl, 10 mM MgCl$_2$, 10 mM KCl, 5% glycerol, 0.01% NP-40, 1 mM DTT, 0.5 mM phenylmethylsulfonyl fluoride, and 1× SIGMAFAST protease inhibitor (Sigma-Aldrich), by shaking in a bead beater (Fritsch) in the presence of glass beads. The lysate was cleared by centrifugation at 18,000 rpm and incubated with IgG Sepharose 6 Fast Flow beads (GE Healthcare) for 120 min at 4 °C. Beads were washed with lysis buffer (without protease inhibitors) and subsequently eluted through TEV cleavage at 16 °C for 120 min. The eluted protein was concentrated and further purified by SEC on a Superdex 200 HiLoad 16/60 column (GE Healthcare) equilibrated with lysis buffer. The respective fractions were pooled and concentrated using an Amicon Ultra-4 cellulose centrifugal unit to a final concentration of ~5.5 mg/ml.

**In vitro binding assays**. To assess in vitro binding, the GST-tagged bait protein (15 μg GST-*Ct*Rsa4-UBL, 15 μg GST-*Ct*Ytm1-UBL or 30 μg GST-*Ct*Kap104) was incubated with 15 μg wild-type *Ct*MIDAS or respective MIDAS loop mutants (as indicated in Fig. 5c, d and Fig. 7a; Supplementary Fig. 6a, b) and/or 140 μg *Ct*Rea1-NAAA[+] ring (amino acid residues 1–2390) and incubated in binding buffer (50 mM Tris, pH 7.5, containing 80 mM NaCl, 5 mM MgCl$_2$, 5% Glycerol, 2% DMSO, 2 mM Na$_2$SO$_4$, 0.01% NP-40, and 1 mM DTT) at 4 °C for 45 min. Nucleotides (ATP or AMPPNP, Sigma-Aldrich) or Rbin-1 (Axon Medchem) where added to a final concentration of 2 and 0.1 mM, respectively. Next, GST-tagged bait proteins and bound material were pulled-down by incubation with 70 μl GSH–agarose (Prontino, Macherey-Nagel) in Mobicol columns (MoBiTec), at 4 °C for 45 min. The flow-through was discarded and beads were washed five times with 500 μl binding buffer. Subsequently, elution was performed with the addition of 50 μl elution buffer (binding buffer containing 30 mM reduced GSH). Eluates were analyzed by sodium dodecyl sulphate (SDS) polyacrylamide gel electrophoresis on 4–12% polyacrylamide gels (NuPAGE, Invitrogen) and Coomassie staining.

**Affinity purifications from yeast lysate**. Aid-HA-*REA1* strains transformed with plasmid-based constructs were grown over night in synthetic dextrose complete medium lacking leucine (SDC-Leu). Exponentially growing cells were shifted to YPD rich medium for an additionally 6–7 h and depletion of Aid-HA-Rea1 was induced 2 hours before cell harvesting by the addition of auxin (500 μM final concentration)[60]. For the ATP release assay *RIX1-TAP L3*-Flag strains transformed with the respective plasmids were grown in synthetic raffinose complete medium lacking leucine (SRC-Leu) and cells were shifted for 6 h to YPG medium before harvesting.

Cells were resuspended in lysis buffer (50 mM Tris-HCl, pH 7.5, 100 mM NaCl, 1.5 mM MgCl$_2$, 5% glycerol, 0.1% NP-40 and 1 mM DTT) and lysed in a bead beater (Fritsch) in the presence of glass beads. The lysate was cleared by centrifugation at 4000 rpm for 10 min at 4 °C and subsequent centrifugation at 17,500 rpm for 25 min at 4 °C. Next, IgG Sepharose 6 Fast Flow (GE Healthcare) was added to the clear lysate and the mixture was incubated for 90 min on a turning wheel at 4 °C. The IgG beads were washed with lysis buffer, resuspended in 500 μl lysis buffer containing TEV protease and incubated for 90 min at 16 °C. The eluate was incubated with Anti-Flag M2 Affinity Gel (Sigma-Aldrich) for 1 h at 4 °C. Beads were washed with lysis buffer and samples were eluted by the addition of lysis buffer supplemented with 0.15 mg/ml Flag peptide (Sigma-Aldrich). The samples were precipitated by the addition of TCA (10% final concentration) and

resuspended in SDS sample buffer. The ATP release assay shown in Fig. 6g was performed as described[23].

**Antibodies**. The following antibodies were used for western blot analyses: anti-Nog1 antibody (1:5000) and anti-Nsa2 antibody (1:10,000), provided by Micheline Fromont-Racine, anti-Rsa4 antibody (1:10,000), provided by Miguel Remacha, anti-Rpl3 antibody (1:5000), provided by Jonathan Warner, horseradish-conjugated anti-Flag antibody (1:15,000, Sigma-Aldrich A8592), anti-CBP antibody (1:50,000, Thermo Scientific, CAB1001), secondary horseradish-peroxidase-conjugated goat anti-rabbit antibody (1:2000, Bio-Rad-170-6515).

**Yeast two-hybrid analysis**. Plasmids expressing the bait proteins, fused to the *GAL4* DNA-binding domain (GAL4-BD), and the prey proteins, fused to the *GAL4* activation domain (GAL4-AD), were co-transformed into the reporter strain PJ69-4A[61]. Yeast two-hybrid interactions were documented by spotting representative transformants in tenfold serial dilution steps on SDC-Trp-Leu and SDC-Trp-Leu-His (*HIS3* reporter gene) plates.

**Fluorescence microscopy**. Living yeast cells expressing GFP-tagged proteins were grown to the logarithmic growth phase and imaged by fluorescence microscopy using a Zeiss Imager Z1 microscope.

**Reporting summary**. Further information on research design is available in the Nature Research Reporting Summary linked to this article.

## Data availability
The source data underlying Figs. 4c, 5c, 5d, 6d, 6g, and 7a and Supplementary Fig. 6a, 6b are provided as a Source Data file. The atomic coordinates and structure factors for the reported crystal structures have been deposited in the Protein Data Bank (PDB) under accession codes 6QT8 MIDAS, 6QTA MIDAS/RSA4, and 6QTB MIDAS/YTM1. Other data are available from the corresponding authors upon request.

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

## Acknowledgements

We thank Jürgen Kopp, Claudia Siegmann, and Gabi Müller from the BZH/Cluster of Excellence: CellNetworks crystallization platform for protein crystallization, Satyavati Kharde and Matthias Becker for data collection of the various iodine datasets, Léonie Strömich for excellent technical assistance during reconstitution of the MIDAS–Ytm1-UBL complex, and ESRF for support and access to beamlines. We are grateful to Dieter Kressler for providing plasmids, to Gunter Stier for initial cloning, and to M. Fromont-Racine, M. Remacha, and J. Warner for sharing antibodies. This work was supported by grants from DFG Hu363/15-1 (to E.H.) and the Leibniz program (SI 586/6-1 to I.S.). I.S.

and E.H. are investigators of the Cluster of Excellence: CellNetworks and acknowledge support through EcTOP1.

## Author contributions

M.T., Y.L.A., V.M., I.S., and E.H. conceived the study. Y.L.A. crystallized proteins and Y.L.A. and I.S. determined the crystal structures. Experiments were performed and analyzed by M.T. and V.M. The Aid-HA-Rea1 strain and plasmids were generated by M.T., excluding the PY-NLS mutant constructs and the Ytm1/Rsa4-UBL expression constructs, which were generated by V.M. and Y.L.A., respectively. M.T., V.M., Y.L.A., I.S., and E.H. wrote the paper. All authors discussed the results and commented on the paper.

## Additional information

**Competing interests:** The authors declare no competing interests.

**Peer Review Information:** *Nature Communications* thanks the anonymous reviewer(s) for their contribution to the peer review of this work. Peer reviewer reports are available.

