## [Peer Review File · Nature Communications]

Reviewers' comments:

Reviewer #1 (Remarks to the Author):

Ahmed et al. describe a series of experiments to characterize the interactions between the MIDAS domain of Midasin, a large AAA+ ATPase and its two targets during 60S maturation, the UBL domains of the Rsa4 and Ytm1 biogenesis factors. Structural models of the two MIDAS-UBL interactions, obtained from X-ray crystallographic experiments, backed up by extensive in vivo mutagenesis studies, extends our knowledge of structural basis of Midasin-directed factor removal during maturation of the 60S ribosomal subunit. The data is presented in a clear and logical sequence, teasing out the precise functions of three sequence elements (including an extensive characterization of the NLS in element II) that set these interactions apart from MIDAS-Ligand interactions found in integrin receptors. In particular, the study nicely complements the recent cryo-EM studies of full-length Midasin to offer new insights into how this molecular machine is able to remove factors from maturing ribosomes. The functional separation of the subcellular localization and AAA+ ring binding functions of element II will be particularly useful for further studies of Midasin function.

A few points:

General: While the structural work involves MIDAS interactions with both Ytm1 and Rsa4, the subsequent validation work focuses solely on Rsa4. While one can rightly deduce that many of the interactions will be similar, more information and probing of the MIDAS-Ytm1-UBL interaction as well as a discussion of similarities/differences would be welcome and should be included (i.e. are the residues mutated in Rsa4 also conserved in Ytm1?). Surely the experiments in Fig. 7a were also performed with the Ytm1-UBL domain. What did they show? One of the interesting aspects of Midasin function is that the two factors it removes are in different regions of the pre-60S. Thus comparing and contrasting the two ligand interactions would inform the function of Midasin in biogenesis.

1. Regarding element III, the authors focus on the importance of this element in binding UBL domains. Their two-hybrid data (Fig. 3c and d) suggests a strong reduction in binding to Rsa4, but the in vivo phenotypes (Fig. 3e) are far less pronounced, with even the double mutation not showing a phenotype at 30°C. However, element III rearranges independently of UBL interactions when the MIDAS domain engages the AAA+ ring (Fig. 7b), suggesting that the loop repositioning may occur before UBL domain binding. The authors should de-emphasize the role of the loop in ligand binding (i.e. the word "crucial" at the end of the third paragraph of results), as the bigger role of the beta hairpin may be to increase the affinity of the MIDAS domain for the AAA+ ring.

2. In the binding studies (Fig. 7a), did the authors consider using the Midasin AAA2 insert deletion? If the idea that the AAA2 insert and the MIDAS domain both engage the ring center, it would bolster the argument that AA2 insert rearrangement is necessary for MIDAS (and substrate UBL) binding.

3. In my printout, the segmented densities shown in Figure 7c are hard to see. The idea that element II threads through the center of the AAA+ ring is perhaps the most exciting hypothesis to emerge from this study and a larger, more informative figure should be made to illustrate this point.

I recommend publication of a revised manuscript.

Reviewer #2 (Remarks to the Author):

Summary:

In this manuscript, Ahmed et al utilize x-ray crystallography to study the C-terminal MIDAS domain of the large AAA-ATPase Rea1/Midasin, which is required for ribosome production. Earlier work from the Hurt lab established that the MIDAS domain of Rea1 specifically interacts with the UBL domains of assembly factors Ytm1 and Rsa4 and that this MIDAS-UBL interaction is critical for Rea1 driven assembly factor release. Moreover, the Hurt lab previously established that the MIDAS-UBL interactions mimic integrin-receptor interfaces through conserved metal-ion binding residues from both the MIDAS and UBL domains. In this manuscript, the authors present three structures of the Rea1 MIDAS domain including an apo structure and complexes with the UBL domains from both Ytm1 and Rsa4. As expected the Rea1 MIDAS domain resembles the structures of MIDAS domains from integrins, however there are three elements in Rea1 not found in canonical MIDAS domains (this was already inferred from sequence alignments described initially in Garbarino and Gibbons, 2002). Upon binding to the UBL domains, element III from the MIDAS domain becomes an ordered beta-hairpin that provides an additional binding interface for the UBL domains. This is a surprising finding and one that would have been impossible to determine in the absence of MIDAS-UBL complex crystal structures. The authors complement their structures with a series of elegant *in vivo* and *in vitro* experiments to tease apart the function of each Rea1 MIDAS specific element. First, they show that element I is critical for Rea1 function *in vivo* and Rsa4 binding. Next, they show that element III from the MIDAS domain is important for Rsa4 and Ytm1 binding by Y2H and they demonstrate the significance of element III *in vivo* with yeast complementation assays. Finally, they characterize element II/loop and find that element II is essential for Rea1 function and nuclear localization. Analysis of the well-conserved element II sequence suggests that it harbors an NLS sequence, which is confirmed by immunofluorescence experiments with GFP-tagged Rea1 constructs. Beyond nuclear import the authors also demonstrate that element II is important for association with the Rea1-AAA ring by *in vitro* pull downs. The authors speculate that element II could insert into the AAA-ring and be the trigger for Rsa4/Ytm1 release, however there is currently no experimental evidence for this.

Major Concern:

Overall this work is well presented and supported, and a nice complement to the recent cryo-EM structures of Rea1 and earlier studies from the Hurt lab. My only significant concern is whether or not this manuscript represents a big enough advance to warrant publication in Nature Communications. This work addresses detailed information about a small domain from a very large protein that has already been extensively characterized. While new and important information was learned about the Rea1 MIDAS domain it is the opinion of the reviewer that this manuscript is lacking in overall novelty.

Minor Concerns:

1. The title of the manuscript is ambiguous. The authors should at a minimum include Rea1 in the title because readers may not make the connection between "integrin receptor-ligand-type complex" and Rea1/Midasin.
2. The authors show no experimental electron density maps. The authors should include non-biased density maps that illustrate the density of important elements from their structures such as the metal-ion binding site and element III (beta-hairpin).
3. The authors solved the structure of the MIDAS domain by SAD-phasing with iodine, which would suggest there are well-ordered iodine binding site(s) within the MIDAS domain. The authors should mention the iodine binding site(s) in the main text or methods section.
4. I found the Element II vs delta-loop nomenclature confusing. The authors should pick one name

and stick with it and not use them interchangeably throughout the manuscript.

5. Is the CT Rea1 N1AAA construct capable of hydrolyzing ATP? Do the MIDAS and/or UBL domains stimulate hydrolysis?

6. Figure 3

(a) Element III should be colored red

(b) The residues numbers in the inset should be labeled in red so it's clear these are the CT residues.

(f) The docking of the MIDAS structure into the cryo-EM map is very nice but it's impossible to see the quality of the fit from this zoomed out view.

7. Figure 7

(a) Still looks like there is weak binding to the AAA-ring in the absence of element II. To ensure ATP doesn't influence binding the pull downs with the delta loop should be done in the presence and absence of ATP and non-hydrolysable analogues.

(b) In-consistent color scheme for Midas apo and delta loop with Figures 2 and 3

Point-by-point response to the reviewers' comments

Reviewer #1 (Remarks to the Author):

Ahmed et al. describe a series of experiments to characterize the interactions between the MIDAS domain of Midasin, a large AAA+ ATPase and its two targets during 60S maturation, the UBL domains of the Rsa4 and Ytm1 biogenesis factors. Structural models of the two MIDAS-UBL interactions, obtained from X-ray crystallographic experiments, backed up by extensive *in vivo* mutagenesis studies, extends our knowledge of structural basis of Midasin-directed factor removal during maturation of the 60S ribosomal subunit. The data is presented in a clear and logical sequence, teasing out the precise functions of three sequence elements (including an extensive characterization of the NLS in element II) that set these interactions apart from MIDAS-Ligand interactions found in integrin receptors. In particular, the study nicely complements the recent cryo-EM studies of full-length Midasin to offer new insights into how this molecular machine is able to remove factors from maturing ribosomes. The functional separation of the subcellular localization and AAA+ ring binding functions of element II will be particularly useful for further studies of Midasin function.

A few points:

General: While the structural work involves MIDAS interactions with both Ytm1 and Rsa4, the subsequent validation work focuses solely on Rsa4. While one can rightly deduce that many of the interactions will be similar, more information and probing of the MIDAS-Ytm1-UBL interaction as well as a discussion of similarities/differences would be welcome and should be included (i.e. are the residues mutated in Rsa4 also conserved in Ytm1?). Surely the experiments in Fig.7a were also performed with the Ytm1-UBL domain. What did they show? One of the interesting aspects of Midasin function is that the two factors it removes are in different regions of the pre-60S. Thus comparing and contrasting the two ligand interactions would inform the function of Midasin in biogenesis.

As suggested by this reviewer, we extended our analysis regarding the MIDAS-Ytm1-UBL interaction. Specifically, we performed binding assays between the C_{Ytm1}-UBL, C_{MIDAS} wild-type, C_{MIDAS} Δ loop and the C_{Rea1} NAAA+ ring, and included this new data in revised Supplementary Fig. 6a. Moreover, we performed genetic interaction studies between the Rea1 element III mutants and conditional *rsa4* and *ytm1* mutants, respectively, revealing the functional importance of the β -hairpin for both MIDAS-ligand interactions (see also next comment).

1. Regarding element III, the authors focus on the importance of this element in binding UBL domains. Their two-hybrid data (Fig. 3c and d) suggests a strong reduction in binding to Rsa4, but the *in vivo* phenotypes (Fig. 3e) are far less pronounced, with even the double mutation not showing a phenotype at 30°C. However, element III rearranges independently of UBL interactions when the MIDAS domain engages the AAA+ ring (Fig. 7b), suggesting that the loop repositioning may occur before UBL domain binding. The authors should de-emphasize the role of the loop in ligand binding (i.e. the word “crucial” at the end of the third paragraph of results), as the bigger role of the beta hairpin may be to increase the affinity of the MIDAS domain for the AAA+ ring.

We de-emphasized this part of the manuscript and no longer use the word “crucial” to describe the role of element III. However, element III has a clear *in vivo* role that could be

shown by additional experiments, which were requested by this reviewer, revealing a strong genetic relationship between the Rea1 element III mutants and previously described *rsa4* or *ytm1* mutants (Ulbrich C, et al., 2009, Cell and Bassler J et al., 2010, Mol Cell.). Specifically, the combination of *rea1* Y4859A I4871A or Y4859R I4871R with either the *rsa4-1* or *ytm1* S78L mutant alleles induced synthetic lethality. This finding underscores the importance of MIDAS element III for ribosome biogenesis and suggests a similar mechanism for the MIDAS-Rsa4-UBL and MIDAS-Ytm1-UBL interaction, respectively.

2. In the binding studies (Fig. 7a), did the authors consider using the Midasin AAA2 insert deletion? If the idea that the AAA2 insert and the MIDAS domain both engage the ring center, it would bolster the argument that AA2 insert rearrangement is necessary for MIDAS (and substrate UBL) binding.

We performed these binding assays with a *CtRea1* NAAA+ construct lacking the Helix 2 insertion in AAA2; however, we did not observe an increased interaction of Rea1 Δ AAA2 insert with the *CtRea1*-MIDAS::Rsa4-UBL heterodimer (actually it appears that the interaction could be slightly decreased, but we did not follow this systematically). We show the outcome of this experiment to this reviewer, and also mention this finding in the text (data not shown). Due to this additional data, we revised this part in the discussion accordingly.

Binding assay with a *CtRea1* NAAA+ construct lacking the Helix 2 insertion in AAA2 compared to the normal *CtRea1* NAAA+ construct. Note that the recruitment of the *CtMIDAS*-*CtRsa4*-UBL was not increased.

3. In my printout, the segmented densities shown in Figure 7c are hard to see. The idea that element II threads through the center of the AAA+ ring is perhaps the most exciting hypothesis to emerge from this study and a larger, more informative figure should be made to illustrate this point.

We have revised this figure as suggested, by providing a cutaway view of the *SpRea1* NAAA+ ring model and the cryo-EM density of the MIDAS domain and the unassigned density within the NAAA+ pore.

I recommend publication of a revised manuscript.

Reviewer #2 (Remarks to the Author):

Summary:

In this manuscript, Ahmed et al utilize x-ray crystallography to study the C-terminal MIDAS domain of the large AAA-ATPase Rea1/Midasin, which is required for ribosome production. Earlier work from the Hurt lab established that the MIDAS domain of Rea1 specifically interacts with the UBL domains of assembly factors Ytm1 and Rsa4 and that this MIDAS-UBL interaction is critical for Rea1 driven assembly factor release. Moreover, the Hurt lab previously established that the MIDAS-UBL interactions mimic integrin-receptor interfaces through conserved metal-ion binding residues from both the MIDAS and UBL domains. In this manuscript, the authors present three structures of the Rea1 MIDAS domain including an apo structure and complexes with the UBL domains from both Ytm1 and Rsa4. As expected the Rea1 MIDAS domain resembles the structures of MIDAS domains from integrins, however there are three elements in Rea1 not found in canonical MIDAS domains (this was already inferred from sequence alignments described initially in Garbarino and Gibbons, 2002). Upon binding to the UBL domains, element III from the MIDAS domain becomes an ordered beta-hairpin that provides an additional binding interface for the UBL domains. This is a surprising finding and one that would have been impossible to determine in the absence of MIDAS-UBL complex crystal structures. The authors complement their structures with a series of elegant in vivo and in vitro experiments to tease apart the function of each Rea1 MIDAS specific element. First, they show that element I is critical for Rea1 function in vivo and Rsa4 binding. Next, they show that element III from the MIDAS domain is important for Rsa4 and Ytm1 binding by Y2H and they demonstrate the significance of element III in vivo with yeast complementation assays. Finally, they characterize element II/loop and find that element II is essential for Rea1 function and nuclear localization. Analysis of the well-conserved element II sequence suggests that it harbors an NLS sequence, which is confirmed by immunofluorescence experiments with GFP-tagged Rea1 constructs. Beyond nuclear import the authors also demonstrate that element II is important for association with the Rea1-AAA ring by in vitro pull downs. The authors speculate that element II could insert into the AAA-ring and be the trigger for Rsa4/Ytm1 release, however there is currently no experimental evidence for this.

Major Concern:

Overall this work is well presented and supported, and a nice complement to the recent cryo-EM structures of Rea1 and earlier studies from the Hurt lab. My only significant concern is whether or not this manuscript represents a big enough advance to warrant publication in Nature Communications. This work addresses detailed information about a small domain from a very large protein that has already been extensively characterized. While new and important information was learned about the Rea1 MIDAS domain it is the opinion of the reviewer that this manuscript is lacking in overall novelty.

The recent cryo-EM structures allowed the structural characterization of the gigantic Rea1 protein, but its MIDAS domain and its interaction with the Rea1 AAA+ ring was only poorly resolved in both publications. Even though the MIDAS domain is small in comparison to the rest of the protein, it performs key functions in eukaryotic ribosome assembly, which was not understood at a structural level to date. Our study changed this view and provided important

novel structural and functional insights regarding the Rea1-MIDAS::UBL-ligand interaction. Thus, we believe that our findings will help to further unravel the mechanisms how this unusual AAA+ ATPase drives eukaryotic ribosome assembly. Moreover, these Rea1-MIDAS::ligand complexes have a strong structural similarity to integrin α -subunit domains when bound to extracellular matrix ligands, which for integrin biology is a key determinant for force-bearing cell–cell adhesion. However, the presence of additional motifs equips the Rea1-MIDAS for its tasks in ribosome maturation, which are all novel and unexpected findings.

Minor Concerns:

1. The title of the manuscript is ambiguous. The authors should at a minimum include Rea1 in the title because readers may not make the connection between “integrin receptor-ligand-type complex” and Rea1/Midasin.

We have changed the title accordingly.

2. The authors show no experimental electron density maps. The authors should include non-biased density maps that illustrate the density of important elements from their structures such as the metal-ion binding site and element III (beta-hairpin).

We have included experimental electron density maps for the metal-ion binding site and the β -hairpin of the of the CtRea1-MIDAS Δ loop–CtRsa4-UBL structure (Supplementary Fig. 5).

3. The authors solved the structure of the MIDAS domain by SAD-phasing with iodine, which would suggest there are well-ordered iodine binding site(s) within the MIDAS domain. The authors should mention the iodine binding site(s) in the main text or methods section.

Three iodine sites were located, with occupancies of 1.0, 0.69 and 0.29. The first two sites are located along the (unformed) beta-hairpin, while the third site is between β 3 and α 5. We included the information in the method section.

4. I found the Element II vs delta-loop nomenclature confusing. The authors should pick one name and stick with it and not use them interchangeably throughout the manuscript.

We now use the term ' Δ loop' throughout the manuscript.

5. Is the CT Rea1 NAAA construct capable of hydrolyzing ATP? Do the MIDAS and/or UBL domains stimulate hydrolysis?

The CtRea1 NAAA+ construct is capable to hydrolyze ATP (V.M. and E.H., unpublished results; the outcome of this assay is depicted below). Nevertheless, the addition of the CtMIDAS domain or CtRsa4-UBL domain does not influence the ATP hydrolysis in our experimental set up.

In vitro ATPase assay showing the relative ATPase activities of the Rea1 NAAA+ ring construct alone and upon addition of the depicted proteins (left panels). The right panels show the background signals obtained of the depicted factors or buffer in absence of the Rea1 NAAA+ ring.

6. Figure 3

(a) Element III should be colored red

We changed the figure as requested.

(b) The residues numbers in the inset should be labeled in red so it's clear these are the CT residues.

We have changed the colour of the residue numbers to red.

(f) The docking of the MIDAS structure into the cryo-EM map is very nice but it's impossible to see the quality of the fit from this zoomed out view.

We included an additional view in the revised manuscript.

7. Figure 7

(a) Still looks like there is weak binding to the AAA-ring in the absence of element II. To ensure ATP doesn't influence binding the pull downs with the delta loop should be done in the presence and absence of ATP and non-hydrolysable analogues.

We performed these experiments as requested. Binding of the MIDAS Δloop construct is not influenced by the presence of ATP or AMPPNP. The results are now included in the revised Fig. 7a.

(b) In-consistent color scheme for Midas apo and delta loop with Figures 2 and 3

We changed the colours to be consistent with Figure 2 and 3.

REVIEWERS' COMMENTS:

Reviewer #1 (Remarks to the Author):

My concerns have been adequately addresses and I recommend publication.

Reviewer #2 (Remarks to the Author):

Thank you for the opportunity to review this manuscript. This is a very well written manuscript describing new and exciting findings about the MIDAS domain of the large AAA-ATPase Rea1. The work is convincing and well supported by several crystal structures and a combination of in vivo and in vitro studies. The authors have addressed all of my concerns. I think this paper is acceptable for publication as is in Nature Communications.